# Self-Assessment of Competence and Referral Behavior for Musculoskeletal Injections among Dutch General Practitioners

**DOI:** 10.3390/jcm9061880

**Published:** 2020-06-16

**Authors:** Emely Spruit, Marianne F. Mol, P. Koen Bos, Sita M.A. Bierma-Zeinstra, Patrick Krastman, Jos Runhaar

**Affiliations:** 1Department of General Practice, Erasmus MC, University Medical Center, 3000 CA Rotterdam, The Netherlands; e.spruit@erasmusmc.nl (E.S.); marianne.f.mol@gmail.com (M.F.M.); s.bierma-zeinstra@erasmusmc.nl (S.M.A.B.-Z.); p.krastman@erasmusmc.nl (P.K.); 2Department of Orthopaedic Surgery, Erasmus MC, University Medical Center, 3000 CA Rotterdam, The Netherlands; p.k.bos@erasmusmc.nl

**Keywords:** musculoskeletal disorders, therapeutic injections, competence

## Abstract

General practitioners (GPs) are qualified and trained to administer therapeutic musculoskeletal injections when indicated. However, it is unknown to what extend Dutch GPs feel competent to administer these injections in clinical practice. Reluctance among GPs to inject might lead to unnecessary and costly referral to secondary care. An online and offline questionnaire was spread among Dutch GPs, querying demographics, GPs’ self-assessment of injection competence, the number of administered/referred injections and management strategy for musculoskeletal injections. A total of 355 GPs responded. In total, 81% of the GPs considered themselves competent in administering musculoskeletal injections. Self-assessed incompetent GPs performed less injections the last month than self-assessed competent GPs (1.2 ± 1.4 vs 4.8 ± 4.6 injections, P < 0.001). Additionally, they referred four times more often to a colleague GP (0.4 ± 1.0 vs 0.1 ± 0.6 injections per month, P < 0.001) and twice as often to secondary care (1.0 ± 1.3 vs 0.5 ± 0.9 injections per month, P = 0.001). Self-assessed incompetence was associated with female sex (OR [95% CI] = 4.94 [2.39, 10.21]) and part-time work (OR [95% CI] = 2.58 [1.43, 4.66]). The most frequently addressed barriers were a lack of confidence in injection skills, lack of practical training, and uncertainty about the effectiveness and diagnosis of musculoskeletal injections. Although most GPs considered themselves competent to administer musculoskeletal injections, the referral rate to secondary care for several injections was strikingly high. To decrease secondary care referrals, addressing some of the most frequently indicated barriers is highly recommended.

## 1. Introduction

Musculoskeletal problems are common in general practice. In the Netherlands, yearly 700 consultations per thousand registered patients concern problems of the musculoskeletal system [1]. The costs of productivity loss and the burden of disease due to musculoskeletal disorders are high [2,3,4]. Moreover, the prevalence of musculoskeletal diseases is increasing [5]. Therefore, effective diagnostic assessment and treatment of these disorders is of paramount importance. Conservative therapies such as physiotherapy and analgesics are the first treatments of choice [6]. Unfortunately, painkillers are not always effective and non-steroidal anti-inflammatory drugs and opioids have numerous side effects and contraindications [7].

Therapeutic injections are an option when non-drug therapies and painkillers fail or are not recommended. In primary care, musculoskeletal injections normally consist of corticosteroids with or without the addition of local analgesics. The indications for musculoskeletal injections in primary care are diverse. Intra-articular injections are, for example, applied for patients diagnosed with osteoarthritis or as an addition in frozen shoulder treatment [6]. Furthermore, soft tissue injections are used, among other conditions, for carpal tunnel syndrome, de Quervain‘s tendinitis, subacromial bursitis and trigger finger [6]. The long-term effectiveness of musculoskeletal injections is often questioned. [8] Nevertheless, there is evidence that musculoskeletal injections do have short-term effects and are therefore advocated in multiple primary care guidelines [6,9,10,11]. These injections have few adverse effects and, if administered adequately, rarely lead to complications. [9] However, the correct use of musculoskeletal injections for the treatment of common disorders in primary care requires competence and self-confidence by the doctor that administrates these injections.

In Northern Ireland and England, research has been done on the administration of musculoskeletal injections by general practitioners (GPs) [12,13]. These studies concluded that the majority of GPs performed most musculoskeletal injections themselves, rather than referring to a colleague or to secondary care [12,13]. Among Northern Irish and English GPs, female GPs, urban GPs and part-time working GPs were less likely to perform musculoskeletal injections [12,13]. Reported barriers for administering injections were little confidence in injection skills and difficulty in maintaining skills, leading to over-referral to secondary care [12,13,14].

There are few data on the number of musculoskeletal injections by Dutch GPs. A Dutch study on septic arthritis following intra-articular injections demonstrated higher administration of injections by medical specialists (524 injections by 12 specialists) compared to GPs (170 injections by 23 GPs) during a follow-up period [15]. This study, together with the indicated Northern Irish and English studies, give the impression that there is a variation in self-confidence among GPs to perform musculoskeletal injections. We hypothesized that self-assessed incompetent GPs perform less injections and refer more often to secondary care. Moreover, we hypothesize that the female sex, working in an urban practice, working part-time, not being a GP trainer or specialized in the musculoskeletal system would be associated to self-assessed incompetence. Through questionnaires among Dutch GPs, we aimed to answer the following research questions:To what extent do Dutch GPs feel competent in administering musculoskeletal injections?How does self-assessed (in)competence affect their clinical treatment and referral behavior?Which factors are associated with self-assessed incompetence?Which barriers and facilitators to administer musculoskeletal injections do Dutch GPs experience?

## 2. Methods

### 2.1. Development of the Questionnaire

A cross-sectional survey through a self-administered online or paper questionnaire, inspired by the survey of Liddell et al. [13], was developed for Dutch GPs (see Appendix A). The questionnaire included questions on demographic data (sex, organization, work setting, full-time equivalent (FTE), patient population, number of years since the completion of GP training, being a GP trainer and specialization in musculoskeletal disease), the number of musculoskeletal injections performed during the past month and the management strategy (injection by GP self, referral to other GP, or referral to secondary care) for a set of 18 selected musculoskeletal diseases for which an injection is indicated or optional in primary care (inspired by Liddell et al. [13]).

To assess competence, the GPs were asked to what extent they agreed with the statement ‘I consider myself competent in performing musculoskeletal injections’. The outcome was measured using a five-point Likert scale with answer options ‘completely disagree’, ‘disagree’, ‘neutral’, ‘agree’ and ‘completely agree’. Additionally, the questionnaire contained questions about the experienced barriers for musculoskeletal injections and possible ways to overcome these barriers (facilitators). A pilot among three GPs and two GPs in training was performed to make sure the questions were unequivocal.

### 2.2. Study Population and Recruitment

To draw valid conclusions, we estimated a total of 200 completed questionnaires would be necessary, based on previous research [12,13]. Data were collected during the period from 12 November 2018 to 14 January 2019. Through HAweb (an online platform for Dutch GPs), a link to the questionnaire was spread among all 12378 members. Reminders were posted 1, 2.5 and 4 weeks after the first post. Additionally, a paper version of the questionnaire was sent by mail to all 636 GPs in the Haaglanden and Amsterdam–Amstelland regions (randomly selected regions in the Netherlands). Furthermore, the link to the questionnaire was spread using Twitter and Facebook. Finally, forty-two GP trainers who are affiliated with our department were asked to fill out the questionnaire. The total size of the targeted population was 12,378, as all targeted GPs are registered on HAweb.

All GPs participated voluntarily in this study. Ethical approval to conduct the study was not necessary.

### 2.3. Statistical Analysis

To examine whether the sample of responding GPs was representative of the total Dutch GP population, the demographic data were compared to the data of the total Dutch GP population (by eyeballing), obtained through the Netherlands Institute for Health Services Research (NIVEL) [16].

The normality of the questionnaire data was checked using the Kolmogorov–Smirnov test. Using descriptive statistics, the frequency of competent/incompetent GPs was calculated. To compare the referral behavior of competent and incompetent GPs, Mann–Whitney U and chi-squared tests were used to test for the significance of differences between groups on the average number of (referred) injections and the percentage of referrals per indication.

Demographic factors were compared between competent and incompetent GPs, first using chi-squared tests and subsequently with a multivariable logistic regression analysis containing all significant factors. To examine the magnitude of the associations between the competence groups, the number of injections administered and those referred, a linear regression was used, adjusted for significant factors from the multivariable model. Lastly, descriptive statistics were used to demonstrate the frequencies of barriers and facilitators to inject. All data were analyzed using SPSS version 24. The significance level in the analysis was set at P < 0.05.

## 3. Results

### 3.1. Responses

A total of 355 returned questionnaires were analyzed. Fifty-four questionnaires were incompletely filled in. All completed questions from the incomplete questionnaires were used in the analyses.

### 3.2. Representativeness

Sex and work setting were representative of Dutch GPs (Table 1). The other demographic characteristics differed slightly; less full-timers and more recently graduated GPs were included in our study.

### 3.3. Competence

The distribution of the answers to the competence question was 2.0% completely disagree, 5.4% disagree, 11.8% neutral, 57.7% agree and 23.1% completely agree. After completion of GP training (which includes injection training), all GPs are considered capable of injecting. Therefore, ‘neutral’ was considered as incompetent and categories were defined by combining ‘completely disagree’, ‘disagree’ and ‘neutral’ as ‘incompetent’ versus ‘agree’ and ‘completely agree’ as ‘competent’, resulting in 80.8% of the GPs considering themselves competent. 

### 3.4. Number of Injections

#### 3.4.1. Injections Aministered

GPs (N = 339) performed an average of 4.1 (SD = 4.4) musculoskeletal injections during the past month. Forty-four GPs did not administer any injections in this period. The number of administered injections differed significantly between competent and incompetent GPs (4.8 ± 4.6 vs. 1.2 ± 1.4, P < 0.001).

#### 3.4.2. Injections Referred to GP Colleague

On average, 0.2 (SD = 0.7) injections were referred to a GP colleague in the past month. A significant difference was observed between the competence groups; 0.1 (SD = 0.6) injections were referred to a GP colleague in the past month by the competent GPs and 0.4 (SD = 1.0) by the incompetent GPs (P < 0.001).

#### 3.4.3. Injections Referred to Scondary Care

The mean number of injections that were referred to the secondary care in the past month was 0.6 (SD = 1.0). GPs who considered themselves competent referred 0.5 (SD = 0.9) injections to secondary care in the past month, compared to 1.0 (SD = 1.3) referred injections by incompetent GPs (P = 0.001).

### 3.5. Injection Indications

Table 2 shows the percentages of GPs who would refer to secondary care for an injection for different indications. The most referred injection indications were ankle osteoarthritis and metacarpophalangeal (MCP)/proximal interphalangeal (PIP)/distal interphalangeal (DIP)/carpometacarpal (CMC) osteoarthritis. GPs referred the least often for an injection of shoulder bursitis. For eleven of the eighteen injection indications, a significantly higher percentage referred to secondary care among the incompetent GPs compared to the competent GPs.

### 3.6. Factors Associated with Incompetence and Referral Behavior

Table 3 shows the differences in demographic between the competence groups. In the multivariable analysis, factors associated with incompetence were female sex (OR [95% CI] = 4.94 [2.39, 10.21]) and part-time work (low FTE) (OR [95% CI] = 2.58 [1.43, 4.66]). Given the association with sex, stratified regression analyses were done. In the sex-stratified analysis, FTE differed significantly between male competent and incompetent GPs and years since completion of GP training differed significantly among female GPs. Male and female GPs who considered themselves competent administered a significantly higher number of injections than their incompetent counterparts (B [95% CI] = 4.76 [1.21, 8.30] for men and B [95% CI] = 2.01 [1.31, 2.70] for women). The differences in referrals to a colleague GP or to secondary care between male competent and incompetent GPs were not significant (respectively, B [95% CI] = −0.37 [−0.89, 0.14], B [95% CI] = −0.08 [−0.68, 0.52]), while female competent GPs referred significantly less than female incompetent GPs (respectively, B [95% CI] = −0.31 [−0.54, −0.09], B [95% CI] = −0.55 [−0.89, −0.21]).

### 3.7. Barriers

The most common barriers to perform musculoskeletal injections experienced by GPs were a lack of practical training, a lack of confidence in skills, a lack of confidence in diagnosis and uncertainty about the effectiveness of the injection (Figure 1). Few GPs had concerns about medicolegal issues or had a bad experience with injections due to complications. Overall, 28% of the general practitioners indicated that they did not experience any barriers to perform musculoskeletal injections. When analyzed for men and women separately, 45% of the male and 16% of the female GPs did not experience any barriers. Moreover, strikingly more female than male GPs reported a lack of confidence in their skills (50% vs. 14%).

### 3.8. Facilitators

In total, 41% of the GPs considered training in musculoskeletal injection by a rheumatologist or an orthopedic surgeon a possible facilitator (Figure 2). In addition, training by an experienced GP colleague and the possibility to perform the injection intramuscularly instead of intra-articular were often indicated. Overall, facilitators slightly differed between the sexes. Only the option to administer an injection intramuscularly instead of intra-articular showed a clear difference between male and female GPs (respectively, 25% and 49%).

## 4. Discussion

This study demonstrated that one in five GPs considered themselves incompetent in performing musculoskeletal injections. Self-assessed incompetent GPs referred twice as many injections to secondary care than self-assessed competent GPs. The main barriers for GPs to perform joint and soft tissue injections were a lack of practical training and a lack of confidence in their own skills.

A greater percentage of female GPs considered themselves incompetent. In general, women tend to underestimate their skills more often, while men tend to overestimate themselves [17,18]. Looking at the actual performance between the sexes, multiple studies confirmed that women and men are equally skilled, or women even outperform men [19,20,21,22,23]. In the Netherlands, the proportion of female GPs has increased from 36% in 2007 to 51% in 2016 [16]. Therefore, it is important that women consider themselves competent in administering musculoskeletal injections to decrease referrals to secondary care. According to Sharp et al., GPs’ perception of practical skills competence can be increased by performing more procedures [24]. For this reason, injection training for insecure female GPs could be recommended.

Part-time work was also associated with self-assessed incompetence, independent of female sex. Since the hours worked per Dutch GP is decreasing, more GP are working part-time [25]. To keep up with all clinical skills can be hard for a GP, especially when the skills are not used regularly [26]. Part-time workers will encounter musculoskeletal injections less frequently. To prevent unnecessary and costly referrals to secondary care, keeping up with injection skills should be a priority for GPs. Otherwise, referral to colleague GPs (e.g., those specialized in musculoskeletal disorders) should be facilitated.

### 4.1. Comparison with Other Studies

#### 4.1.1. Number of Injections

Liddell et al. concluded that GPs carried out a median number of 17.0 musculoskeletal injections in the last year [13]. The number of injections administered by Dutch GPs was higher, with a mean number of 4.1 injections per month. It is possible that English guidelines recommend a musculoskeletal injection less often compared to Dutch GP guidelines or that English GPs have more/stronger personal barriers to injecting. Unfortunately, Liddell et al. and Gormley et al. did not examine referrals to secondary care [12,13].

In accordance with previous studies, we found that female GPs performed significantly fewer injections than male GPs [12,13]. Since these studies did not question self-assessed competence, it is not possible to compare the competence in musculoskeletal injections between English and Dutch GPs [12,13].

#### 4.1.2. Barriers

In agreement with previous studies, we found that GPs’ most frequently reported barriers to carry out musculoskeletal injections were a lack of confidence in skills and a lack of practical training [12,13]. In contrast, uncertainty about the effectiveness of injections was a common barrier among Dutch GPs [12,13]. This is noteworthy, as the Dutch College of General Practitioners (NHG) provides guidelines with evidence-based recommendations regarding musculoskeletal injections. Despite this, GPs indicated to question these NHG guidelines, though they are usually well adhered to these guidelines [6,27].

#### 4.1.3. Training

English GPs preferred training on patients, but this facilitator was the least popular among Dutch GPs [13]. Previously, training on patients was deemed superior to training on a phantom for improving confidence in performing musculoskeletal injections [12]. However, training on phantoms is more feasible, because the training could be given in large groups and GPs can easily practice the injection multiple times consecutively.

### 4.2. Limitations

First of all, there is a possibility of non-response bias in our study. In particular, more GPs who have a special interest in injections or in the musculoskeletal system could have completed the questionnaire. In the Netherlands, there is a dedicated group of general practitioners who specialize in musculoskeletal disease. Colleague GPs can refer patients to them or consult them with questions about musculoskeletal pathology. As expected, the prevalence of these specialized GPs in our study was higher than among the entire Dutch GP population. This might have led to an overestimation of self-assessed competence in the current study. This overestimation could be even larger, as self-assessed incompetent GPs might feel reluctant to fill out the questionnaire (truthfully), despite the fact that the survey was anonymous.

Our study included slightly less full-timers when compared to Dutch primary care. As mentioned earlier, part-time work was associated with self-assessed incompetence. Therefore, the percentage of competent GPs could be underestimated.

Moreover, it is unclear whether GPs are capable of adequate self-assessment of their injection skills. In a study among GPs, Janssen et al. found that there was only a moderate correlation between self-assessed competence of technical skills and actual competence (as measured with a performance-based test) [28].

Furthermore, the lack of information collected on working conditions is a limitation as well. The level of colleague support or the psychosocial work environment may also influence the decision to refer injections to a colleague.

Finally, we decided to dichotomize GPs’ competence by combining the answer categories ‘completely disagree’, ‘disagree’ and ‘neutral’ versus ‘agree’ and ‘completely agree’. Obviously, when ‘neutral’ was added to the definition of competence, more GPs consider themselves competent (92.7%), but this does not change the significance of any of the outcomes.

### 4.3. Implications for Practice and Further Research

Although the self-assessed competence among GPs was high, many patients were referred to secondary care for musculoskeletal injections by incompetent GPs. To decrease referrals to secondary care, more training on injection skills is recommended, both in GP training and as refresher courses. NHG-led courses already exist for injections for shoulder, de Quervain’s tenosynovitis, carpal tunnel syndrome, trigger finger/thumb, knee and trochanter pain syndrome [6]. Referral to specialized GPs instead of to secondary care seems a good option for incompetent GPs. Another possibility is that orthopedists visit primary care to assess and treat patients together with the general practitioner.

In recent years, the effectiveness of intramuscular injections for musculoskeletal disease, compared to traditional intra-articular injections, has been studied. If an intramuscular injection is not inferior to an intra-articular injection, it may be easier for GPs to carry out musculoskeletal injections in the future, as injecting intramuscularly was considered a facilitator by GPs to perform musculoskeletal injections. Furthermore, the clinical effectiveness of intramuscular injections has already been shown for hip osteoarthritis [29] and the effectiveness of an intra-articular injection versus intramuscular injection in patients with rotator cuff disease showed no significant differences between the methods of administration [30]. Recently, a randomized controlled trial on the non-inferiority of an intramuscular injection in comparison with an intra-articular injection in patients with knee osteoarthritis finished data collection [31].

Finally, uncertainty about the effectiveness of musculoskeletal injections was a common barrier among GPs in the current study. A more in-depth analysis on the reasons why GPs question the effectiveness of musculoskeletal injections would be of interest.

## Figures and Tables

**Figure 1 jcm-09-01880-f001:**
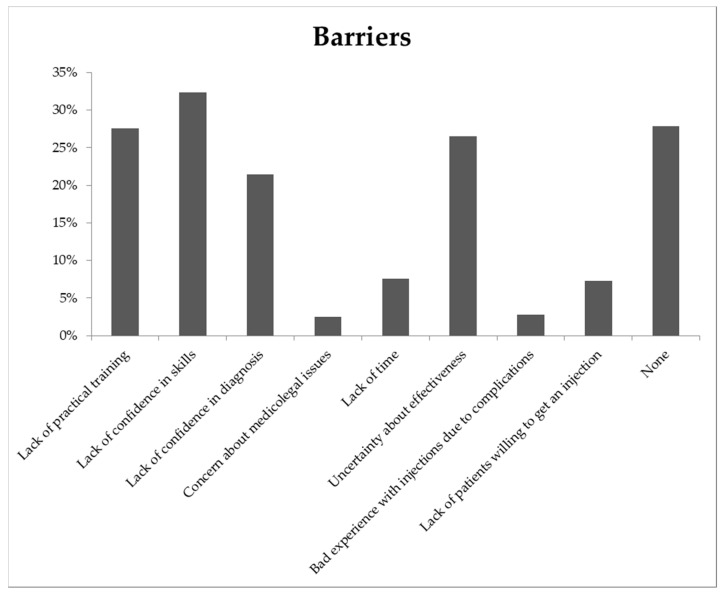
Percentage of GPs reporting selected barriers to performing musculoskeletal injections (N = 355).

**Figure 2 jcm-09-01880-f002:**
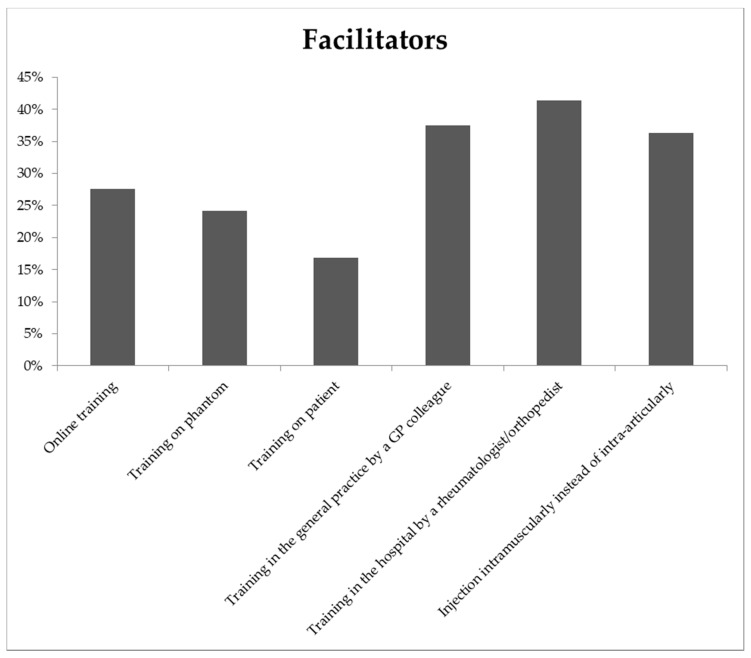
Percentage of GPs reporting selected facilitators to performing musculoskeletal injections (N = 355).

**Table 1 jcm-09-01880-t001:** Frequencies of demographics of the study and Netherlands Institute for Health Services Research (NIVEL).

Demographic		Current Study % ^a^	NIVEL %
Sex	Man	42.0	49 ^b^
	Woman	52.1	51 ^b^
	Neutral	0.3	-
Organization	Solo practice	19.2	17.8 ^b^
	Duo practice	23.7	40.3 ^b^
	Group practice	20.8	41.9 ^b^
	Health center	15.5	-
	Other ^c^	11.5	-
Work setting	Rural	26.8	30.2 ^d^
	Urban	67.6	69.8 ^d^
FTE	0–0.20	1.7	0.2
	0.21–0.40	5.9	2.8
	0.41–0.60	20.6	15.9
	0.61–0.80	30.7	28.5
	0.81–1	35.2	52.6
Patient population ^e^	< standard practice	13.2	-
	= standard practice	25.9	-
	> standard practice	55.2	-
Completion GP training	<5 years	23.1	13.6 ^f^
	5–15 years	33.2	29.4 ^f^
	16–25 years	18.9	18.5 ^fg^
	>25 years	19.4	38.6 ^fg^
Trainer	Yes	24.5	-
	No	70.1	-
GP musculoskeletal system	Yes	1.7	-
	No	93.0	-

^a^ Percentages do not add up to 100% due to missing data. ^b^ Data calculated on regular established general practitioners. ^c^ Other organizations include observers, nursing home, non-practicing etc. ^d^ Rural is defined as little urban and not urban. Urban is defined as moderately urban, strong urban and very strong urban. ^e^ Standard practice = 2095 patients. ^f^ Data calculated on the basis of Table 2 of the NIVEL brochure: the number of general practitioners who have graduated in the Netherlands for each graduation year or graduation period for their status as at 1 January 2016. ^g^ Data concern the period 16–26 years instead of 16–25 years and the period 27–42 years instead of > 25 years in connection with specified graduation periods instead of graduation year.

**Table 2 jcm-09-01880-t002:** Percentages of general practitioners (GPs) who referred an injection to secondary care per indication.

Indication	All GPs % (N)	Competent % (N)	Incompetent % (N)	*P*-Value
Indications for which an injection is recommended by the Dutch College of General Practitioners				
Carpal tunnel syndrome	36.1 (330)	32.2 (267)	52.4 (63)	0.003
Knee osteoarthritis	26.7 (333)	23.0 (270)	42.9 (63)	0.001
Plantar fasciitis	22.7 (326)	20.9 (263)	30.2 (63)	0.116
De Quervain’s tenosynovitis	22.5 (329)	16.1 (267)	50.0 (62)	0.000
Supraspinatus tendinitis	20.0 (320)	16.9 (260)	33.3 (60)	0.004
Trigger finger/thumb	17.1 (333)	12.6 (270)	36.5 (63)	0.000
Osteoarthritis shoulder	16.9 (325)	14.5 (262)	27.0 (63)	0.018
Trochanteric bursitis	5.8 (330)	4.1 (268)	12.9 (62)	0.007
Bursitis shoulder	2.1 (333)	0.7 (270)	7.9 (63)	0.000
Indications for which an injection is not recommended by the Dutch College of General Practitioners				
Ankle osteoarthritis	50.0 (324)	50.2 (261)	49.2 (63)	0.888
Osteoarthritis MCP/PIP/DIP	48.9 (321)	44.2 (258)	68.3 (63)	0.001
Osteoarthritis CMC	48.9 (319)	45.1 (257)	64.5 (62)	0.006
Sacroiliitis	44.4 (420)	43.4 (258)	48.4 (62)	0.479
Achilles tendinitis	33.1 (323)	31.2 (260)	41.3 (63)	0.126
Prepatellar bursitis	11.9 (327)	9.1 (264)	23.8 (63)	0.001
Medial epicondylitis	8.5 (329)	7.5 (266)	12.7 (63)	0.185
Lateral epicondylitis	7.0 (328)	6.4 (265)	9.5 (63)	0.385
Olecranon bursitis	5.8 (327)	4.9 (265)	9.7 (62)	0.148

**Table 3 jcm-09-01880-t003:** Percentages of self-assessed (in)competent GPs in performing injections per demographic group.

Demographic		N	Competent %	Incompetent %	*P*-Value
Sex	Man	149	93.3	6.7	-
	Woman	185	70.8	29.2	0.000
Organization	Solo practice	70	81.4	18.6	-
	Duo practice	85	78.8	21.2	-
	Group practice	74	79.7	20.3	-
	Health center	55	83.6	16.4	-
	Other	41	80.5	19.5	0.967
Work setting	Rural	95	78.9	21.1	-
	Urban	240	81.2	18.8	0.631
Full-time equivalent (FTE)	0–0.20	6	66.7	33.3	-
	0.21–0.40	21	61.9	38.1	-
	0.41–0.60	73	68.5	31.5	-
	0.61–0.80	109	84.4	15.6	-
	0.81–1	125	80.2	19.2	0.001
Patient population	< standard practice	47	76.6	23.4	-
	= standard practice	92	83.7	16.3	-
	> Standard practice	196	80.1	19.9	0.584
Completion GP training	< 5 years	82	81.7	18.3	-
	5–15 years	118	78.8	21.2	-
	16–25 years	67	73.1	26.9	-
	> 25 years	69	89.9	10.1	0.090
Trainer	Yes	87	81.6	18.4	-
	No	249	80.3	19.7	0.793

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
