# Peer review of "Self-Assessment of Competence and Referral Behavior for Musculoskeletal Injections among Dutch General Practitioners"

_jcm, 2020, doi:10.3390/jcm9061880_

Round 1
Reviewer 1 Report
Review of manuscript jcm-824610
General appreciation
This study sought to evaluate the self-assessed competence of Dutch GPs in administering musculoskeletal injections and the relation of perceived competence to referral behaviour. It also sought to evaluate factors associated with self-assessed competence level and the barriers to carrying out musculoskeletal injections, which in turn could be acted upon to improve competence and reduce the need for referral. The study rationale is well explained. This research yields new and important knowledge that could be used to improve Dutch medical practice and reduce healthcare costs. Nevertheless, some improvements could be made by addressing the elements below.
Abstract
- Most Dutch GPs consider themselves competent to administer musculoskeletal injections, yet referral to secondary care for several injections was found to be quite high. The authors recommend more training. In light of some of the other barriers to performing injections identified in the study, can the authors mention that these barriers may also need to be addressed (e.g. uncertainty about injection effectiveness)?
Introduction
- The authors state that the prevalence of musculoskeletal disease is increasing (line 37), but cite an inappropriate reference to support this claim: the cited study (Jordan et al., 2010) reports annual prevalence rates for the year 2006 only. It is recommended to either remove the claim or replace the reference.
- In lines 70-75, there are five research questions. Can these be separated out for more clarity?
- Can the authors state their hypotheses and predictions regarding the association between GP self-assessed competence level and number of injections or referrals, as well as their hypotheses and predictions regarding some of the sociodemographic factors investigated?
- Can the research question in lines 74-75 be re-worded? As currently formulated, it assumes interventions are needed even prior to conducting the study. For example: “Which factors associated with self-assessed incompetence and which barriers to carrying out injections can be acted upon to improve GP competence and reduce the need for referral?”
Methods
- Adjusting for sex in regression analyses has important limitations because the variable “sex” is often a proxy for unmeasured work and non-work-related factors that can influence injection and referral behaviour [men and women with the same job title can experience different work situations, organizational constraints (e.g. high work intensity and lack of time), and psychosocial work environment, e.g. colleague support]. Can the authors perform sex-stratified regression analyses in order to identify relevant factors associated with injection and referral behaviour in men and in women separately? Can the same be done with respect to their results on barriers and facilitators to performing injections? It would be interesting to see if the relative importance of barriers and facilitators changes when analyzing men and women separately. If sample size does not permit stratification for regression analyses, can the authors state this and address the limits of adjusting regression analyses for sex in their discussion? See for example Silverstein et al., 2009 (Gender adjustment or stratification in discerning upper extremity musculoskeletal disorder risk? Scand J Work Environ Health 2009;35(2):113–126).
- On the topic of sex and gender, the authors appropriately include a third sex/gender category in their questionnaire, as an implicit acknowledgement that individuals who do not fit into the male and female categories should be included in health research. However, they seem to use the terms “sex” and “gender” interchangeably, even though they are not synonymous: the study questionnaire asks “What is your gender?” (based on google translate), but in the methods (line 79) and in Table 1, the label “sex” is used. Care should be taken to use these terms appropriately and consistently throughout the manuscript, depending on what is meant. See for example https://cihr-irsc.gc.ca/e/47830.html#d1 and https://orwh.od.nih.gov/sex-gender.
- Can the sub-title “spread” (line 93) be changed to something more reflective of the content of that paragraph, for example, “study population and recruitment”?
- Can the authors clarify the total size of the targeted population? Are the 636 GPs in the Haaglanden and Amsterdam-Amstelland and the 42 GP trainers all included in the 12,378 members (lines 97-101)? This will affect the calculation of the survey response rate.
- Can the authors specify the study design in their methods? For example, “cross-sectional survey through self-administered online or paper questionnaire”.
- Were efforts made to increase response rate through reminders? If so, this should be specified in methods.
- Can the paragraph describing regression analyses be re-worded (lines 112 to 117) to remove the emphasis placed on uncovering statistical significance? Emphasis should instead be placed on estimating the magnitude of hypothesized associations between self-assessed competence level and referral/injection behaviour.
- Please add a statement regarding physician consent to participate in the study and a statement about ethical approval by a relevant institution to conduct the study.
Results
- Did the authors examine and can they comment on the interrelationships between their sociodemographic factors? For instance, how did rural/urban setting relate to solo/duo/group practice/health centre and to the patient population size and was this different for male and female GPs?
- Did the authors have information on the age and ethnicity of their GP population, and how did it compare to the NIVEL data?
- Can sub-title 3.4.1. be changed as “self-administered” implies that the GPs administered the injection to themselves. For example, “Injections administered”.
- Can sub-title 3.6. be changed to reflect not only the results on factors associated with self-assessed incompetence but also results of the regression analyses on the relations between self-assessed competence and injection or referral behaviour?
- Suggestion to change Figure 1 title to “Percentage of GPs reporting selected barriers to performing musculoskeletal injections”.
- Suggestion to change Figure 2 title to “Percentage of GPs citing selected facilitators to performing musculoskeletal injections”.
Discussion
- Line 198: remove “in a representative group” since the sample differed from the general Dutch GP population in terms of full-time workers and amount of training, as stated by the authors.
- Other limitations that may be worth noting include the lack of information collected on working conditions that might affect injection and referral behaviour. Part-time work and lack of time were included, but the level of colleague support or other elements of work organization or the psychosocial work environment may also influence the decision to refer injection to a colleague (availability of supplies?).
References
- Please correct reference #11 (incomplete - not possible to retrieve) and reference #17 (issue with the date).
Reviewer 2 Report
Thank you for leaving me the opportunity to read a well-structured and well-written paper.
The topic is important in the perspective of health service management.
Your final statement is perhaps the most interesting question for answer - who are the right patients to offer injections? I would encourage your group to continue researching this topic before offering injection techniques training to all GPs.
